# Foliar Spraying of Glycine Betaine Alleviated Growth Inhibition, Photoinhibition, and Oxidative Stress in Pepper (*Capsicum annuum* L.) Seedlings under Low Temperatures Combined with Low Light

**DOI:** 10.3390/plants12132563

**Published:** 2023-07-06

**Authors:** Nenghui Li, Kaiguo Pu, Dongxia Ding, Yan Yang, Tianhang Niu, Jing Li, Jianming Xie

**Affiliations:** College of Horticulture, Gansu Agricultural University, Yingmen Village, Anning District, Lanzhou 730070, China; linh@st.gsau.edu.cn (N.L.); pukaiguo1998@163.com (K.P.); dingdongxia100741@outlook.com (D.D.); yy@st.gsau.edu.cn (Y.Y.); niutianhang@outlook.com (T.N.)

**Keywords:** pepper, low temperature combined with low light, glycine betaine, antioxidant, photoinhibition, biological membrane, subcellular localization

## Abstract

Low temperature combined with low light (LL stress) is a typical environmental stress that limits peppers’ productivity, yield, and quality in northwestern China. Glycine betaine (GB), an osmoregulatory substance, has increasingly valuable effects on plant stress resistance. In this study, pepper seedlings were treated with different concentrations of GB under LL stress, and 20 mM of GB was the best treatment. To further explore the mechanism of GB in response to LL stress, four treatments, including CK (normal temperature and light, 28/18 °C, 300 μmol m^−2^ s^−1^), CB (normal temperature and light + 20 mM GB), LL (10/5 °C, 100 μmol m^−2^ s^−1^), and LB (10/5 °C, 100 μmol m^−2^ s^−1^ + 20 mM GB), were investigated in terms of pepper growth, biomass accumulation, photosynthetic capacity, expression levels of encoded proteins *Capsb*, cell membrane permeability, antioxidant enzyme gene expression and activity, and subcellular localization. The results showed that the pre-spraying of GB under LL stress significantly alleviated the growth inhibition of pepper seedlings; increased plant height by 4.64%; increased root activity by 63.53%; and decreased photoinhibition by increasing the chlorophyll content; upregulating the expression levels of encoded proteins *Capsb A*, *Capsb B*, *Capsb C*, *Capsb D*, *Capsb S*, *Capsb P1*, and *Capsb P2* by 30.29%, 36.69%, 18.81%, 30.05%, 9.01%, 6.21%, and 16.45%, respectively; enhancing the fluorescence intensity (OJIP curves), the photochemical efficiency (Fv/Fm, Fv′/Fm′), qP, and NPQ; improving the light energy distribution of PSΠ (Y(II), Y(NPQ), and Y(NO)); and increasing the photochemical reaction fraction and reduced heat dissipation, thereby increasing plant height by 4.64% and shoot bioaccumulation by 13.55%. The pre-spraying of GB under LL stress also upregulated the gene expression of *CaSOD*, *CaPOD*, and *CaCAT*; increased the activity of the ROS-scavenging ability in the pepper leaves; and coordinately increased the SOD activity in the mitochondria, the POD activity in the mitochondria, chloroplasts, and cytosol, and the CAT activity in the cytosol, which improved the LL resistance of the pepper plants by reducing excess H_2_O_2_, O_2_^−^, MDA, and soluble protein levels in the leaf cells, leading to reduced biological membrane damage. Overall, pre-spraying with GB effectively alleviated the negative effects of LL stress in pepper seedlings.

## 1. Introduction

Plants live in an ever-changing environment that is often unfavorable for growth and development or stressful, which can include biotic stress (pathogen infection) and abiotic stresses such as drought, high and low temperature combined with low light, nutrient deficiency, and excessive salt or toxic metals in the soil [1]. Among these, low temperature combined with low-light stress is a major environmental factor that limits crop productivity in agriculture, resulting in threatened economic benefits [2]. In addition, low temperature combined with low light inhibits plant biological processes from seed germination to seedling growth, flowering, and fruiting [3,4,5]. It is worth noting that overwintering vegetables are often damaged by low temperatures combined with low light in facilities in northwestern China.

Pepper (*Capsicum annuum* L.), the second largest edible crop of Solanaceae after tomatoes, has an annual yield of more than 37 million tons and great economic benefits [6]. As the most important off-season vegetable cultivated in solar greenhouses in northwestern China, it is often exposed to low temperatures combined with low-light conditions in winter or early spring months. Studies have shown that the low temperatures combined with low-light stress damage the cell membrane, then increase osmotic substances, thereby weakening photosynthesis and destroying leaf tissue [5]. Moreover, antioxidants accumulate [7], key gene expression is downregulated, and antioxidant enzyme activity reduces in pepper leaves [8].

When plants are exposed to stresses, morphology (height, stem diameter, leaf count, and dry biomass) change is the most intuitive outcome of their response to an external stimulus. Similarly, the increase in MDA level and electrolyte leakage rate (EC) typically serves as the main reason for damage-induced biological membrane rigidification [9,10,11]. As a typical indication of photosynthesis, chlorophyll fluorescence parameters such as Fv/Fm, qP, NPQ, and Y(II) provide a detailed insight into the electron transfer process in photosystem II and beyond [12]. Likewise, the OJIP phase of the Chl fluorescence induction curve can be the most useful surrogate value for whole-plant seedling vigor [13]. In addition, the leaf relative water content (RWC), which includes free and bound water content (FWC, BWC), is an indicator of stress and tolerance in spring wheat [14] and cassava [15], and a reduction in RWC is a common consequence of stress [16].

LL stress can cause oxidative damage to plants and produce excessive reactive oxygen species (ROS), such as superoxide anion (O_2_^–^), H_2_O_2_, and hydroxyl (-OH); therefore, plants have evolved enzymatic antioxidant systems, such as superoxide dismutase (SOD), peroxidase (POD), catalase (CAT), and ascorbate peroxidase (APX), to cope with stress, either by stress avoidance or stress tolerance [17]. Under abiotic stress (heavy metals, drugs, drought, temperature extremes, etc.), plants are generally exposed to oxidative stress and produce excess free radicals, therefore, they respond to stress by regulating the activity of enzymes such as SOD, POD, and CAT [5,18,19,20]. In the ascorbate–glutathione cycle, APX reduces H_2_O_2_ by using ascorbate as an electron donor. Hamid et al. found that hormones regulate leaf senescence mainly by modulating H_2_O_2_ through different mechanisms, such as enzymatic antioxidants (SOD, POD, and CAT) and nonenzymatic antioxidant defense systems, under water stress in wheat [21]. Furthermore, the subcellular localization of antioxidant enzymes remains an important indicator of abiotic stress, and studies have shown that organelles (mitochondria, chloroplasts, and cytosol) mitigate stress at extreme temperatures by coordinating the intracellular activity of antioxidant enzymes [22,23].

However, when pepper seedlings experience low temperatures combined with low-light stress, the traditional solution is to install a passive climate control system in the greenhouse; however, the installation costs are high [24]. Therefore, it is imperative to mitigate low temperatures combined with low light by spraying with exogenous osmolytes.

Glycine betaine (GB) has been widely studied as an osmolyte in plants, bacteria, animals, and humans [25]. GB is also a protein stabilizer known as an osmoprotector that plays an important role in osmoregulation and is one of the main nitrogenous compatible osmolytes in Poaceae. When plants encounter abiotic stresses, exogenous GB applications have been shown to induce the expression of genes that are involved in oxidative stress responses that limit the accumulation of ROS and lipid peroxidation in cultured plant cells under drought, salinity, heavy-metal, chilling, and waterlogging conditions, and even stabilize photosynthetic structures under stress.

Although research on GB under abiotic stresses is sufficient, the study of low temperatures combined with low light with exogenous GB in pepper seedlings is still lacking. Therefore, the aim of the present study was to explore some of the physiological functions that regulate plant tolerance, especially in terms of photosynthesis, osmoregulatory substances, and antioxidant enzyme activity, and their ability to coordinate subcellular distribution.

## 2. Results

### 2.1. Changes in Growth, Water Content, and Antioxidant Enzyme Activity under LL Stress by GB at Different Concentrations

In Figure 1A, the pepper seedlings wilted severely (T0) after 24 h of LL (low temperature combined with low light); however, spraying with different concentrations of GB obviously alleviated the wilting. The cotyledons of the pepper in the T1 treatment were pendulous, the tips of the second true leaves were dehydrated and wilted in the T3 treatment, all of the true leaves curled in the T4 treatment, and, interestingly, the leaves in the T2 treatment were under normal conditions. The results of the LL treatment for 7 days regarding the accumulation of dry material in the aboveground part of the seedlings are shown in Figure 1B. At different concentrations of GB, the leaf count, plant height, and stem diameter of the pepper seedlings showed the same trend, which gradually increased from T0 to T2 and decreased from T2 to T4. Among these, the value of T2 was significantly higher than that of the other treatments in terms of the leaf count and stem diameter; however, the plant height was not significantly different from that of the others. The relative intracellular water content of the leaves of the pepper seedlings was also used to measure the degree of damage caused by LL stress. The BWC (bound water content), FWC (free water content), RWC (relative water content), and WSD (water saturation deficit) in Figure 1C show that low temperature combined with low light increased BWC and decreased FWC; however, in T2, with the sparse nature of GB, the BWC was significantly higher than that of T0, and the FWC was significantly lower than that of T0. The RWC was the highest and the WSD was the lowest in T0 compared with the other four treatments; however, T2 was significantly different from T0, T1, T3, and T4. It is also noteworthy that the RWC of T0, T1, T3, and T4 increased by 20.75%, 9.14%, 16.18%, and 11.63%, compared to T2, respectively, while the WSD of T0, T1, T3, and T4 was lower that of T2 by 83.07%, 7.32%, 66.09%, and 47.48%, respectively.

As shown in Figure 1D, the antioxidant enzyme activities of SOD, CAT, and APX were clearly observed in the pepper seedlings in response to LL stress at different GB concentrations. Compared with the other treatments (T0, T1, T3, and T4), the activities of SOD, CAT, and APX in T2 were significantly different and higher. The SOD activity of T2 increased by 368% in T0, 83.86% in T1, 64.51% in T3, and 25.86% in T4; the CAT activity of T2 increased by 20.96% in T0, 40.27% in T1, 22.92% in T3, and 27.43% in T4; and the APX activity of T2 increased by 9.57% in T0, 69.43% in T1, 33.31% in T3, and 60.98% in T4.

In conclusion, exogenous spraying with 20 mM of GB (T2) was the optimal concentration to respond to LL stress in subsequent experiments.

### 2.2. GB Affected the Growth, Dry Biomass, and Root Activity under LL Stress

Figure 2 shows that exogenous GB had no effect on the growth of the pepper seedlings at a normal temperature (CK) after 7 days of treatment; however, compared with LL treatment, exogenous GB treatment (LB) significantly increased the plant height by 4.64%, increased the dry and fresh weight of the pepper shoot by 13.55% and 7.53%, respectivley, and enhanced the root activity by 63.53%. The dry and fresh weight of the pepper root was not significantly different from the treatments with LL.

### 2.3. GB Affected the Photosynthetic Pigment Content and Analysis of the OJIP Curve under LL Stress

In order to understand how GB alleviates the PSΠ photoinhibition induced by LL stress, the photosynthetic pigment content and fluorescence intensity (OJIP curve) were determined for each treatment (Figure 3). At normal temperatures, exogenous GB significantly increased the chla and chlT contents but had no effect on the chlb and car contents. In contrast, under LL stress, the addition of exogenous GB significantly increased the chla, chlb, chlT, and car contents of the pepper leaves by 21.32%, 20.92%, 21.23%, and 20.37%, respectively, compared to the LL treatment. The OJIP curves showed that GB enhanced the fluorescence intensity at the normal temperatures, and LL stress significantly decreased the fluorescence intensity at the J-I-P stage. However, GB significantly increased the fluorescence intensity at LL and was greater than that under LL stress.

### 2.4. GB Affected Chlorophyll Fluorescence Parameters and Energy Distribution under LL Stress

In Figure 4A, Fv/Fm (the maximum photochemical efficiency of PSΠ) and Fv′/Fm′ (the efficiency of excitation energy capture by open PSΠ reaction centers) levels showed the same trend. The pre-spraying of GB under normal conditions effectively increased the Fv/Fm levels, with no difference in Fv′/Fm′ levels, and both levels decreased significantly under LL stress; however, the addition of GB significantly increased the Fv/Fm and Fv′/Fm′ levels, even when there was no difference compared to normal temperature. As seen in Figure 4B, the values of the Y(NO) and Y(NPQ) levels were significantly lower under LL stress, and the addition of GB under LL stress significantly increased the values of Y(NO) and Y(NPQ) and was not different from CK and CB treatments at the normal temperature. The order of the Y(II) levels was CK > CB > LB > LL. As shown in Figure 4C, the NPQ level was significantly reduced by the LL treatment, which was significantly increased the LB treatment and was not different from the CK and CB treatments. Compared with the CK treatment, the qP level was significantly reduced in the CB treatment; however, compared with the LL treatment, its level was significantly increased in the LB treatment and was not different from the CB treatment. A portion of the energy reaction in PSII is shown in Figure 4D. The LL stress resulted in a significant decrease in P and Ex and a significant increase in D. However, the application of GB under LL stress reversed this phenomenon, with P and Ex significantly increasing and D significantly decreasing in the CB treatment.

### 2.5. GB Affected the Expression of Genes Encoding the PSΠ Reaction Center Proteins under LL Stress

Figure 5 shows that the expression levels of Capsb A, Capsb B, Capsb C, Capsb D, Capsb S, Capsb P1, and Capsb P2 genes were significantly downregulated after LL treatment, especially the relative expression of *Capsb S* and *Capsb P1*, with values of 0.8% and 0.29%, respectivley; however, this result was reversed by pre-spraying with GB. Compared with LL treatment, the expression of Capsb A, Capsb B, Capsb C, Capsb D, Capsb S, Capsb P1, and Capsb P2 genes were upregulated at LB, with values of 30.29%, 36.69%, 18.81%, 30.05%, 9.01%, 6.21%, and 16.45%, respectively.

### 2.6. GB Affected the Osmotic Substances under LL Stress

To investigate how GB regulates leaf membrane permeability and osmotic substances in pepper seedlings under LL stress, EC, proline, soluble protein, and endogenous GB contents were measured (Figure 6A–D). There was no difference in the content of EC, proline, or soluble protein in the CK and CB treatments; however, they were significantly increased in the LL treatment, and their levels were significantly lower than those of LL with the addition of exogenous GB. 

In addition, there was no difference in endogenous GB content under CK and LL treatments. However, when exogenous GB was sprayed on the leaves, the GB content was significantly higher in the CB and LB treatments, and the GB content in the LB treatment was significantly lower than that of CB. 

### 2.7. GB Affected the ROS Scavenging-Ability under LL Stress

As shown in Figure 7A, the ROS-scavenging ability of peppers also reflected the vital role of exogenous GB under LL stress. At the normal temperature, the CK and CB treatments showed no difference in the activity of SOD or POD. Compared with the normal temperature, the activities of SOD and POD were significantly decreased under LL stress; however, their activities significantly increased under LB treatment, and the SOD activity of LB was similar to that of CK. For CAT activity, the results showed that the treatment with exogenous GB addition was significantly higher than that of the treatment without GB addition at the normal temperature and LL stress, and CAT activity was higher under the LB treatment than under the CB and CK treatments.

In Figure 7B, the results showed that the expression of CaSOD and CaPOD genes were significantly upregulated by the exogenous pre-spraying of GB under normal conditions, and sharply downregulated under LL stress. However, it is notable that the GB pre-spraying reversed the low level of expression caused by LL stress and significantly upregulated the expression of CaSOD, CaPOD, and CaCAT genes under LB treatment.

Furthermore, H_2_O_2_ and O_2_^−^ contents significantly increased under LL treatment, but significantly decreased after the pre-spraying of GB (LB treatment). Likewise, the shades of color in NBT and DAB histochemical staining showed the levels of H_2_O_2_ and O_2_^−^ (Figure 8C), indicating that the color of LL was darker than that of LB, and the colors of CK and CB were light. These results suggest that exogenous GB enhances the ROS-scavenging ability under LL stress.

### 2.8. GB Affected the Subcellular Localization of Antioxidant Systems in Pepper Leaves under LL Stress

Each subcellular component can coordinate antioxidant protection. Therefore, we examined how the subcellular components coordinate the activities of the antioxidant enzymes SOD, POD, and CAT under the application of exogenous GB under LL stress. In Figure 8A, the data show that, under LL stress, the addition of exogenous GB significantly increased the SOD activity in the mitochondria, whereas it made no difference in the chloroplasts, but significantly decreased the SOD activity in the cytosol. Furthermore, the SOD activity in the chloroplasts and cytosol was significantly higher under the low temperature combined with low light treatment compared to that of the normal temperature treatment. Figure 8B shows that the CAT activity in the cytoplasmic solutes was significantly higher than that in the mitochondria and chloroplasts, and there was no significant difference in the pre-spraying with GB at the normal temperature; however, the addition of GB significantly increased the CAT activity in the chloroplasts, mitochondria, and cytosol under LL stress. Moreover, the addition of GB under LL stress significantly increased the activity of POD in the chloroplasts, mitochondria, and cytosol, whereas the difference in the levels of activity at normal temperatures was not extremely significant between the CK and the CB treatments (Figure 8C). In addition, the MDA and soluble protein contents in the chloroplasts, mitochondria, and cytosol were all significantly higher in the LL treatment than in the other treatments (Figure 8C,D).

In conclusion, the addition of exogenous GB can respond to LL stress by coordinately increasing the activity of SOD in mitochondria, increasing the activity of POD and CAT in the chloroplasts, mitochondria, and cytosol, and decreasing the levels of MDA and soluble proteins in the chloroplasts, mitochondria, and cytosol.

## 3. Discussion

Abiotic stresses have a negative impact on crop performance. In particular, LL stress is a major constraint on crop yield and quality [26] and effects plant physiology at the both whole-plant and cellular levels at all stages of developmental, from seedling to senescence [27]. Therein, the seedling stage of plants is sensitive to adverse environmental factors. In this study, it has been shown that LL-stress-induced growth inhibition is associated with decreased photosynthesis and antioxidant enzyme activity in pepper plants. As an osmolyte, GB effectively improves plant stress resistance [28,29], increasing the root activity of pepper seedlings, decreasing photoinhibition, lowering the levels of osmoregulatory substances, increasing the activity of antioxidant enzymes, and coordinating the activity of subcellular antioxidant enzymes.

Morphological indexes are the most intuitive expression of a plant’s response to an external stimulus. Dry biomass, as the morphological establishment trait, is equally sensitive when plants face stress [30]. Meanwhile, root activity affects the ability of the plant to absorb nutrients and, thus, the morphological establishment of the aboveground parts under abiotic stress [31]. Studies have shown that drought stress negatively affects root morphology and reduces photosynthetic pigments (Chl), which severely inhibits the growth and biomass production of tomato seedlings [32]. Likewise, it was found that the exogenous spraying of GB significantly enhanced the growth characteristics, biomass, proteins, and chl content of wheat under chromium (Cr) stress, which was consistent with the results of this research. Exogenous GB increased the shoots’ fresh weight and root activity, thereby increasing the dry biomass of the pepper seedlings under LL stress (Figure 2). Thus, we found that a low temperature of 10/5 °C decreased the root activity of the pepper plants in this experiment, which inhibited the morphological establishment of the aboveground parts by reducing the uptake of nutrients, and that the application of exogenous GB may have increased the nutrient NPK [33], thereby increasing the dry matter accumulation of the pepper seedlings and reducing growth inhibition.

Moreover, LL stress affects photosynthesis in plants. The typical indicators are the photosynthetic pigment (chl) content and chlorophyll fluorescence parameters, which describe the mechanism of photosynthesis and the physiological status of photosynthesis in plants and are considered intrinsic probes for studying the relationship between the photosynthesis of plants and the environment [34]. When high salt concentrations enter plant cells, the membrane system and function of cyst-like bodies in the chloroplasts can be disrupted [35]. Heat stress impairs the energy flux reaching the reaction centers of PSII and the ability of the active reaction centers to photon capture [36], and a low temperature combined with low light significantly reduces Fv/Fm, Y(NO), NPQ, Fv′/Fm′, and [Y(II)] in watermelon plants [37]. Interestingly, one study has reported that cadmium (Cd) stress decreased Fo, Fm, and Fv/Fm levels; however, this was reversed after the foliar application of GB [38]. Another study investigated whether exogenous GB could preserve the photochemical activity of PSII, maintain higher Fv/Fm, and recover more rapidly from photoinhibition under drought stress [39], the results of which were similar to those of this experiment. The pepper seedlings had reduced Fv/Fm, Y(NO), NPQ, Fv′/Fm′, Y(II), and qP contents (Figure 4A–C). Finally, in our study, we examined pepper seedlings under LL stress, which reduced P and Ex and increased D; however, GB resisted the low temperature combined with low-light stress by reducing D and increasing P and Ex (Figure 4D), which was consistent with the study that reported that the energy distribution ratio of the PSII reaction center changed significantly (P and Ex showed an overall downward trend, while D was promoted) under salt stress [40]. Similarly, the OJIP curve, which represents the fluorescence intensity, can indicate the light and efficiency of the plant [41]. Studies have shown that the PSII system of shade-intolerant plants is more vulnerable, with J-P sites having a lower fluorescence intensity in low light [42]. Salt stress has also been found to reduce the intensity of all of the peaks of the plant chlorophyll fluorescence transient (OJIP) curve, indicating a significant deactivation of energy on the donor and acceptor side of the PSII [43]. Again, LL stress reduced the intensity of the J-P dot, but pre-spraying with GB increased the intensity of the J-P dot in the pepper plants under LL treatments (Figure 3). We hypothesize that GB may mitigate the energy deficiency caused by LL stress on the donor and acceptor side of the PSII and respond to LL stress by improving the sufficiency of light energy utilization and reducing the dissipation of excess energy from the pepper seedlings.

Additionally, the *psbA* and *psbD* genes encode the turnover proteins D1 and D2, and the *psbB* and *psbC* genes encode the internal light-harvesting complex proteins CP43 and CP47, which are located in the core of the reaction center, to recover and maintain the stability of the PSII in plants after some stress [44]. The *PsbP* proteins are required for a normal cystoid structure and are also required for the assembly/stability of the PSII complex and photoautotrophy in Arabidopsis [45,46]. The *PsbS* genes are rapidly activated in green organisms by light stress. In this work, pre-spraying with GB significantly upregulated the transcript expression levels of *Capsb A*, *Capsb B*, *Capsb C*, *Capsb D*, *Capsb S*, *Capsb P1*, and *Capsb P2* under LL stress (Figure 5). Therefore, we hypothesize that the enhanced tolerance to LL stress in pepper plants is due to the redistribution of the PSII center energy by GB using the encoded protein *CapsB*, which reduces the dissipation of excess energy and allows more light energy to be captured by the PSII center, thereby increasing the photosynthetic efficiency and accelerating the recovery of the PSII from the photoactivated state, allowing it to recover more quickly from photoinhibition [47].

Osmotic regulatory substances are also indispensable in characterizing abiotic stress. The levels of EC, soluble protein, and proline increased under drought stress in soybean leaves [48] and decreased under low temperature combined with low light in pepper leaves [5]. Remarkably, exogenous GB reversed this condition, and the levels of EC, soluble protein, and proline decreased under drought stress. Additionally, in this study, exogenous GB (LB treatment) was found to attenuate the increase in EC, soluble protein, and proline levels in the pepper leaves under the LL treatment; however, we found that the proline levels were significantly higher in the LB treatment than in the CK and CB treatments at the normal temperature (Figure 6A–C). This may be due to the fact that the exogenous GB responds to LL stress mainly by regulating proline metabolism. As an osmotic regulatory substance, GB can alleviate stress under stress conditions by self-accumulation, and studies have shown that GB acts as an accumulator inSolanaceae, Asteraceae, Convolvulaceae, and Amaranthaceae under salt stress [4,49]. Indeed, the experiment results accumulated with the increase in exogenous GB (Figure 2). Notably, the pepper leaves did not accumulate endogenous GB under LL stress (which is in agreement with previous studies) [50], and, compared to the CB treatment, the GB content significantly decreased under the LB treatment, as shown in Figure 6D. It was assumed that the pepper leaves would respond to LL stress by consuming the exogenous GB, converting it into other substances or redistributing the exogenous GB, and that exogenous GB could alleviate the LL stress by reducing the leaf membrane permeability and the osmotic substance levels (proline and soluble protein) in the pepper leaves under low temperatures combined with low-light conditions.

Plants have evolved a number of enzymatic antioxidant systems to control the production and scavenging of ROS in order to cope with stress and to avoid photooxidative damage, either by stress avoidance or stress tolerance [51]. SOD, POD, and CAT are typical ROS scavenger enzymes that can effectively balance excess H_2_O_2_ and O_2_^−^ in plant cells. Studies have suggested that the inhibition of plant growth by salinity may be related to increased oxidative damage due to the accumulation of ROS, but the application of GB can scavenge excess free radicals by enhancing the ROS enzyme activity. It has also been shown that GB can activate the expression of ROS scavenger genes under abiotic stress. In addition, in this experiment, the pepper seedings accumulated a lot of ROS under LL stress; however, the application of GB upregulated the expression of *CaSOD, CaPOD,* and *CaCAT*, enhanced the activity of the antioxidant enzymes (SOD, POD, and CAT), and enhanced the ability of ROS scavenging (Figure 7). Thus, GB activates the enzymes and genes that scavenge reactive oxygen, and the highly active enzyme scavenges excess ROS under low temperature combined with low-light conditions, improving the LL tolerance of peppers.

Finally, owing to abiotic stress, high levels of reactive oxygen species (ROS) are formed in various organelles, leading to cellular damage. The main organelles are the mitochondria, chloroplasts, and cytosol. The PSI and PSII are the reaction centers in the chloroplast, and controlling the ROS levels in the chloroplast is essential for plant survival under stress. ROS is formed less in the mitochondria than in chloroplasts, which are also sites of photorespiration that are surrounded by cytosol [52,53]. At a normal temperature, plants rely on free radical scavenging systems inside organelles such as chloroplasts and mitochondria to keep intracellular free radicals at low levels and to maintain normal physiological metabolism. However, under LL stress, the plant employs the ROS scavenger enzyme activity system to scavenge free radicals and coordinate ROS homeostasis between the organelles [22]. The POD and CAT activities were the strongest in the cytosol of rhododendron leaves under high-temperature stress, consistent with the findings of the present study, and differences in subcellular localization occurred after exogenous hydrogen high temperatures [23]. Similarly, LL stress mobilizes SOD enzymes in the chloroplasts and cytosol to scavenge free radicals in pepper leaves, while exogenous GB under LL stress reduces cell damage by mobilizing SOD in the mitochondria and POD and CAT in the chloroplasts, mitochondria, and cytosol to scavenge excess free radicals (Figure 8A–C). Moreover, in response to stress, GB regulated the osmotic substances MDA and soluble proteins in various organelles (Figure 8D,E). Therefore, we hypothesized that GB coordinates the activities of the free radical scavenging enzymes in each organelle under LL stress and that each enzyme activity is distributed differently and acts differently in different species.

## 4. Materials and Methods

### 4.1. Experimental Materials and Growth Conditions

Pepper (*Capsicum annuum* L.) seeds of ‘HangJiao No. 2’ (provided by Tianshui Agricultural Science Research Institute) were soaked in heated water (50–55 °C), stirred for 30 min, immersed in 25 °C water for 6 h, and allowed to germinate in an artificial climate chamber at 28 °C in the dark for 4 days. When the radicle reached 1–2 mm, two seeds were placed in a nutrient tray (9 cm × 9 cm) containing seedling substrate, vermiculite, and peat (*v*/*v* 3:1:1) at a temperature of 28 °C/18 °C (day/night), a PPFD of 300 μmol m^−2^ s^−1^, a relative humidity of 65%, and a photoperiod of 12/12 h (day/night) in an artificial climate chamber (Ningbo Southeast Instrument, Ningbo, China). A total of 40 pots were grown for each treatment.

### 4.2. Treatments

#### 4.2.1. Different GB Concentration Treatments

When the sixth true leaf was fully expanded, the seedlings were pretreated with five different concentrations of GB solution (CAS:107-43-7; Shanghai Yuanye Biotechnology Co. Ltd., Shanghai, China), including 0 mmol L^−1^ GB (T0), 10 mmol L^−1^ GB (T1), 20 mmol L^−1^ GB (T2), 40 mmol L^−1^ GB (T3), and 80 mmol L^−1^ GB (T4), and then transferred to an artificial climate chamber for 7 days at a temperature of 10 °C/5 °C (day/night), PPFD of 100 μmol m **^−^**^2^ s **^−^**^1^, relative humidity of 65%, and photoperiod of 12/12 h (day/night). It was a preliminary experiment, which was expanded upon in the following experiments.

#### 4.2.2. Alleviating Effects of Exogenous GB on Pepper Plants under LL Stress

Based on the above parameters, the following four treatments were set when the sixth true leaf of the seedlings was expanded:

Normal temperature and light (CK, 28/18 °C, 300 μmol m^−2^ s^−1^);

Normal temperature and light + GB (CB, 28/18 °C, 300 μmol m^−2^ s^−1^, 20 mM GB);

Low temperature combined with low light (LL, 10/5 °C, 100 μmol m^−2^ s^−1^);

Low temperature combined with low light + GB (LB, 10/5 °C, 100 μmol m^−2^ s^−1^, 20 mM GB).

Before LL stress, the CB and LB treatments were sprayed with GB for five successive days (solution containing 0.01% Tween-80 as a surfactant). The CK and LL treatments were replaced with equal amounts of double-distilled water (containing 0.01% Tween-80) and then placed in an artificial climate chamber for 7 days. The third and fourth leaves were selected for subsequent physiological and biochemical analyses after 7 days at 8 am. The experiments for each treatment were repeated three times.

### 4.3. Determinations

#### 4.3.1. Analysis of Morphology and Root Activity

The plant height, stem diameter, number of leaves, and fresh and dry weights of the shoots and roots were measured at 24 h and 7 days after treatment. A tape measure was used to measure the length of the pepper seedlings from the base to the growing point; the stem thickness was measured using a vernier caliper to measure the thickness of the stem below the cotyledons; and the number of fully development leaves was recorded. Whole seedlings were collected, cleaned with water, dried, and separated into roots and shoots to obtain a fresh weight, and dried to a constant weight at 80 °C to obtain a dry weight.

A total of 0.5 g of root tip was weighted into a 10 mL test tube, 10 mL of an equal mixture of 0.4% TTC solution and phosphate buffer was added to completely immerse the roots in the solution, and it was kept in the dark at 37 °C for 2 h, then, 2 mL of 1 mol/L sulfuric acid was added to stop the reaction. Then, the root tips were removed, blotted dry, ground in a mortar with 4 mL of ethyl acetate and a small amount of quartz sand, filtered, and the absorbance values (OD) of the extracts were measured at 485 nm with a UV-1780 spectrophotometer (Shimadzu, Kyoto, Japan) to calculate the root activity.

#### 4.3.2. Related Water Content of Pepper Leaf Cells

The fresh leaves were washed with steamed water and weighed (FW), soaked in 60% sucrose solution for 6 h, kept in the dark at 4 °C, removed, washed several times, and weighed as saturated water (SW). Then, the leaves were placed in an oven at 70 °C for 48 h, and the dry leaf tissue was weighed as DW.

To determine the relative water content (RWC), the fresh leaves were washed with double-steamed water and weighed (Wf), the weighed fresh leaves were soaked in double-steamed water for 24 h, were weighted as a saturated quality (Wt), and then kept in a drying oven at 70 °C for 48 h (Wd). RWC (%) = (Wf − Wd)/(Wt − Wd) × 100. The free water content (FWC) was evaluated as FWC (%) = (FW − SW)/DW × 100%, bound water content (BWC) (%) =TWC-FWC, total water content (TWC) (%) = Wf − Wd, and water saturation deficit (WSD) (%) = 100% − RWC [54].

#### 4.3.3. Determination of Photosynthetic Pigment Content and Chlorophyll Fluorescence Parameter Determination

##### Photosynthetic Pigment Contents of chla, chlb, chlT, and car

Each sample (0.1 g fresh samples) was transferred to a 20 mL tube immersed in 10 mL 80% acetone and then placed in the dark for about 48 h (shaking every 12 h) until the leaves had turned white. Finally, the absorbance values (OD) of the extracts at 663 nm, 645 nm, and 440 nm were measured using a UV-1780 spectrophotometer (Shimadzu, Japan) to calculate the content of chla, chlb, chlT, and car, respectively [55].

##### Chlorophyll Fluorescence Parameters

After 7 days of low temperature combined with low-light treatment, an Imaging-PAM fluorimeter (Walz, Effeltrich, Germany) was used to measure the indexes. Previously, the whole seedlings were kept in the dark for at least 30 min, and the third young leaf was selected for measurement. Fo (minimum fluorescence yield of dark-adapted leaves) and Fm (maximum fluorescence yield of dark-adapted leaves) were obtained by applying a saturation pulse of 2700 μmol m **^−^**^2^ s **^−^**^1^. Subsequently photochemical light (81 μmol m **^−^**^2^ s **^−^**^1^) was turned on every 0.8 s for 20 s, and photoadaptation was performed for 5 min. The actual photochemical efficiency of photosystem II [Y(II)], maximum photochemical efficiency of photosystem II (Fv/Fm), quantum yield of regulatory energy dissipation of photosystem II [Y (NPQ)], quantum yield of nonregulatory energy dissipation of photosystem II [Y(NO)], and portion of energy reaction in PSII were P = (Fv′/Fm′) × qP, Ex = (1 − qP)/(Fv′/Fm′), D = 1 − (Fv′/Fm′) [35].

##### Measurement of the “OJIP” Curve and the JIP Test

Chlorophyll fluorescence induction kinetics (OJIP curves) were captured using a Plant Efficiency Analyzer (Handy PEA, Hansatech, UK), resulting in light response curves and fitting parameters. The leaves were placed in complete darkness for 30 min and then continuously illuminated with 3000 μmol m^−2^ s^−^^1^ of red light to induce a fast chlorophyll fluorescence curve. Instantaneous fluorescence was recorded from 10 μs to 300 s using an OJIP curve diagram to reflect the details of the PSΠ linear electron transfer. In the OJIP curve, “O” represents the “origin” (minimal fluorescence), “P” represents the “peak” (maximum fluorescence), and “J” and “I” represent the “inflection points” between the “O” and “P” levels. Fo is the fluorescence intensity at the “O” level, while Fm is the intensity at the “P” level, and Fv = (Fm − Fo) is the variable fluorescence [54].

#### 4.3.4. Analyses of Biological Membrane Damage

The membrane permeability was evaluated using relative electrical conductivity (EC) [56]. Leaf discs (0.1 g, 0.6 cm in diameter) were weighed and placed in glass test tubes with 15 mL deionized water and vacuumed for 30 min using a vacuum pump. The tubes were sealed and shaken on a shaking table for 3 h. Then, the initial electric conductivity (L1) was determined with a Ddsj-308f-type conductivity meter (Shanghai INESA Scientific Instrument Co., Ltd., Shanghai, China) after it was set to 25 °C for 10 min. Then, the final electrical conductivity (L2) was measured after the tubes were boiled in a water bath for 15 min and then cooled to 25 °C. The electrical conductivity (L0) of deionized water was used as the blank. The REC was evaluated as REC (%) = (L1 − L0)/(L2 − L0) × 100%.

The malondialdehyde (MDA) content can be used as a representative index to evaluate the degree of membrane lipid peroxidation. The leaf samples (0.3 g) were ground into homogenates on ice with 5% trichloroacetic acid (5 mL). Then, the samples were centrifuged at 3000 r/min for 10 min. The supernatant (2 mL) was mixed with 2 mL of 0.67% thiobarbituric acid and dissolved in 5% trichloroacetic acid. The samples were heated at 100 °C for 30 min and then immediately placed into ice water to cool. Finally, the absorbance of the mixture was measured at 450 nm, 532 nm, and 600 nm using a UV-1780 spectrophotometer (Shimadzu, Japan).

The proline and soluble protein contents were determined according to Huang et al., with some fine adjustments [57]. A total of 0.5 g of the leaves was weighted, 10 mL of 3% sulfosalicylic acid solution was added and quickly ground on ice to a homogenous slurry, filtered, and then 2 mL of filtrate was transferred to a test tube, 2 mL of ninhydrin solution and 2 mL of glacial acetic acid were added, the test tube was placed in boiling water for 1 h, quickly cooled on ice, 4 mL of toluene was added, and it was thoroughly mixed. The determined absorbance at 520 nm was used to calculate the content of the proline using a spectrophotometer.

The leaf sample (0.2 g) was weighed into a precooled mortar, 1.6 mL of 50 mmol/L of precooled phosphate buffer (pH 7.8) was added, and the sample was ground in an ice bath to form a homogenate. It was then transferred to a centrifuge tube at 4 °C and centrifuged at 12,000× *g* for 20 min. The supernatant was the crude protein extract. A total of 100 mL of enzyme solution was taken and added to 2.9 mL of Coomassie brilliant blue G-250 solution. After a reaction time of 2 min, OD595 was measured to calculate the content of soluble protein.

#### 4.3.5. Determination of the ROS-Scavenging Capability

The H_2_O_2_ and O_2_^−^ contents were determined by histochemical staining with nitroblue tetrazolium (NBT) and 3,3-diaminobenzidine (DAB), respectively [51]. The leaves were washed with distilled water to remove extraneous material. Then, the leaves were collected in conical flasks, and each treatment included 4 leaves immersed in either 0.1 mg·mL^−1^ NBT (pH 7.8; 0.2 h) or 1 mg·mL^−1^ DAB (pH 7.0; 24 h) staining solutions at 25 °C in the dark to detect H_2_O_2_ and O_2_**^−^**, respectively. Finally, the samples were decolorized with lactic acid: glycerol: ethanol = 1:1:3 (*v*/*v*) in a boiling water bath for 5–6 min and photographed using an EPSON Expression 11,000 XL color image scanner (WinRHIZO Pro LA2400, Canada).

The leaf tissues (0.5 g) were ground in 5 mL ice-cold 0.05 mol sodium phosphate buffer (containing 5 mM EDTA-Na_2_, 2 mM AsA, and 2% (*w*/*v*) PVP, pH 7.8) using a prechilled mortar and pestle. The homogenate was centrifuged at 12,000 r/min for 20 min at 4 °C. The antioxidant activity of the supernatant was determined using a UV-1780 spectrophotometer (Shimadzu, Japan). The superoxide dismutase (SOD) activity was measured according to the method of Geng et al. The SOD activity was measured using the nitrotetrazolium blue chloride (NBT)-illumination method, the peroxidase (POD) activity was measured using the o-methoxyphenol method, the catalase (CAT) activity was measured by ultraviolet absorption, and the ascorbic acid peroxidase (APX) activity was estimated by monitoring the decline in absorbance at 290 nm, due to the oxidation of AsA (ascorbic acid) oxidation by H_2_O_2_ [23].

#### 4.3.6. Determination of the Subcellular Location of ROS-Scavenging Enzymes

Subcellular antioxidant enzyme activity in pepper leaves was reported by Geng et al., and the subsequently determined numerical by visible spectrophotometric determination [23]. The mitochondria, chloroplasts, and cytoplasmic stroma were separated using differential centrifugation. A total of 10 g of leaf sample was taken, 30 mL of pre-cooled extraction buffer (0.05 mol/L Tris-HCl, 0.35 mmol/L sorbitol, 2 mmol/LEDTA, and 2.5 mmol/LDTT, pH = 7.5) and a little quartz sand were rapidly ground on ice, and then the sample was filtered into 4 layers and centrifuged at 500 g/min for 5 min. The supernatant was centrifuged again (2000 rpm/min, 10 min) and the resulting precipitate was the chloroplast fraction. The supernatant was further centrifuged (12,000 rpm/min, 20 min), and the resulting precipitate was the mitochondrial fraction, and the supernatant was the cytosolic fraction. The chloroplasts and mitochondria were washed once with the extraction solution, centrifuged again, and the precipitate was suspended in 5 mL of extraction buffer. The suspensions were then used to determine the antioxidant enzyme activities in each organelle. The NBT method was used to determine the SOD enzyme activity, the guaiacol method was used to determine the POD enzyme activity, and the UV absorption method was used to determine the CAT enzyme activity.

#### 4.3.7. Content of Glycine Betaine

The sample (0.5 g) was homogenized with 5 mL of 0.05% toluene, shaken at 25 °C for 24 h, and centrifuged at 3000× *g* for 20 min. The supernatant was mixed with 3 mL of 1 M hydrochloric acid and KI-I_2_ solution, mixed thoroughly, and shaken at 0 °C for 90 min. Next, 10 mL of 1,2 dichloroethane was added, and, after the separation of the organic layer, the absorption was measured at 365 nm [58].

#### 4.3.8. Quantitative Polymerase Chain Reaction (PCR) Analysis

The total RNA of the different treatments was extracted using the RNA extraction kit provided by Tengen Biotechnology and then reverse-transcribed with a cDNA synthesis kit provided by TaKaRa, Kusatsu City, Japan, to obtain cDNA. Primer 5 software was used to design all primers (Table 1), which were synthesized by Sangon Biotech. The qRT-PCR was performed on BioRad (CFX96, Hercules, California, USA) using the TransStart Top Green qPCR SuperMix kit (TransStart Green, Beijing, China). The reaction mixture contained 2 µL cDNA, 0.6 µL forward primer, 0.6 µL reverse primer, 10 µL qPCR SuperMix, and 6.8 µL RNase-free ddH_2_O. The reaction conditions were 94 °C for 30 s, 95 °C for 5 s, and 60 °C for 30 s, with a total of 40 cycles. The technique was repeated three times for each sample. Actin was used as the reference gene, and the relative gene expression was calculated by the 2^−ΔΔCt^ method.

### 4.4. Statistical Analysis

The statistical analysis was performed with ANOVA using SPSS software (version 22.0, SPSS Institute Limited., California, USA) and EXCEL (Office, 2019). Significant differences (*p* < 0.05, *p* < 0.01) between the means of the different treatments were assessed using Duncan’s multiple range tests. All figures were plotted using Origin Pro software. 9.0 (Origin Lab Institute, incorporated., California, USA).

## 5. Conclusions

In conclusion, the application of GB could be an effective way to improve the tolerance of pepper seedlings under low temperatures combined with low-light stress, which can involve photosynthetic and antioxidant aspects. The pre-spraying of GB alleviated the growth inhibition caused by LL stress by reducing photoinhibition, as indicated by the increased Fv/Fm and chl contents, as well as [Y(II)], Y(NPQ), Y(NO), NPQ, and qP, regulating the lipid peroxidation of membranes, thereby improving the ability to scavenge excess H_2_O_2_ and O_2_^−^ through the increased activity of antioxidant enzymes and the expression of genes encoding antioxidant enzymes and the PSII reaction center proteins, and coordinating the activities of antioxidant enzymes in the mitochondria, chloroplasts, and cytosol. Overall, this study provides a better understanding of the mechanisms by which GB affects the growth of pepper plants under LL conditions and provides preliminary ideas for studying the response mechanisms of pepper seedlings under LL stress.

## Figures and Tables

**Figure 1 plants-12-02563-f001:**
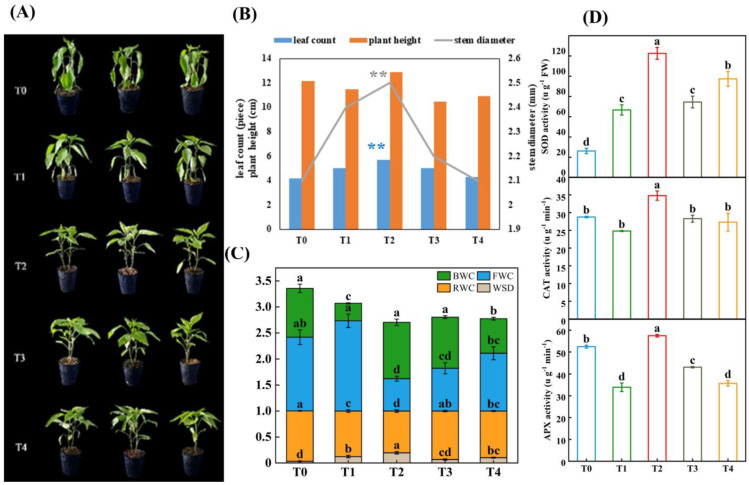
The growth and physiological indices affected by LL stress in pepper seedlings. Photographs and data of pepper plants under low-temperature conditions were obtained after 7 days. (**A**) Morphological changes. (**B**) Accumulation of dry material. (**C**) Related water content, water saturation deficit, free water content, and bound water content. (**D**) Enzyme activity contained SOD, CAT and APX. The results show the mean ± SE of three replicates, and the different letters denote the significant difference among the treatments (*p* < 0.05), according to Duncan’s multiple tests. T0, control; T1, 10 mM GB; T2, 20 mM GB; T3, 40 mM GB; and T4, 80 mM GB. Data are means ± SDs (n = 3). Different lowercase letters represent significant differences (*p* ≤ 0.05) in the same period among treatments, according to Duncan’s test. The “**” in blue and gray color represented leaf number and stem diameter under T2 treatment were significantly different with T0, T1, T3 and T4.

**Figure 2 plants-12-02563-f002:**
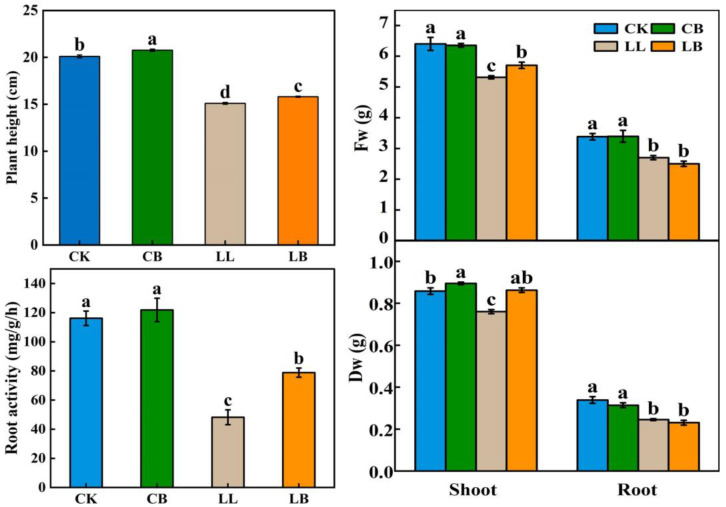
Impact of exogenously pre-spraying with GB on plant height, fresh and dry weight, and root activity of pepper leaves under LL stress. Photographs and data of the pepper plants under LL conditions were obtained after 7 days, which contained plant morphology, dry and fresh weight of the shoot and root, and root activity. CK: Normal conditions, spring RO water. CB: Normal conditions, spring 20 mM GB. LL: Low temperature combined with low light condition, spring RO water. LB: Low temperature condition, spring 20 mM GB. Different lowercase letters represent significant differences (*p* ≤ 0.05) in the same period among treatments, according to Duncan’s test.

**Figure 3 plants-12-02563-f003:**
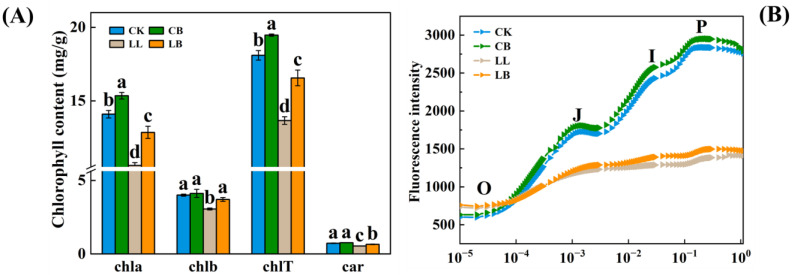
GB affected the photosynthetic pigments content and fluorescence intensity (OJIP curve) by LL stress in pepper seedlings. Data of pepper plants under LL conditions were obtained after 7 days. (**A**) chlorophyll content, contained chla, chlb, chlT (chla + b), and car (total carotenoids) content. (**B**) changes in OJIP curve. Data are means ± SDs (n = 3). Different lowercase letters represent significant differences (*p* ≤ 0.05) in the same period among treatments, according to Duncan’s test.

**Figure 4 plants-12-02563-f004:**
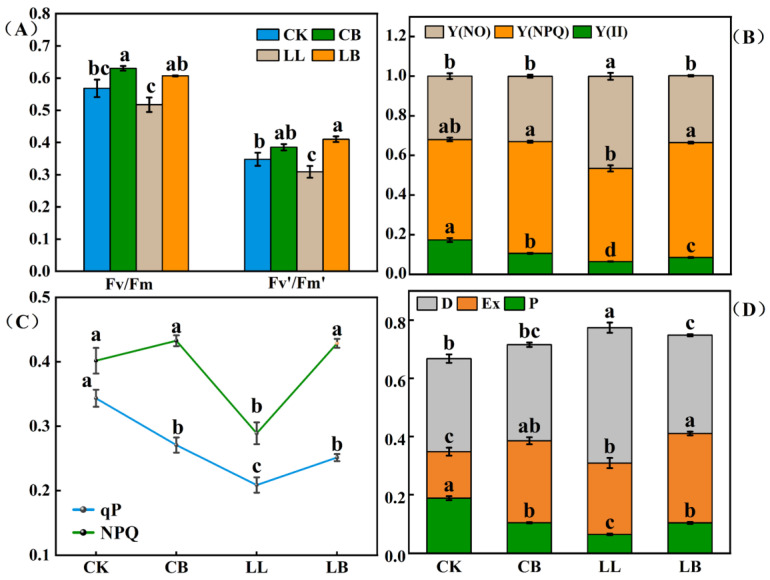
GB affected the chlorophyll fluorescence parameters and light energy distribution of PSΠ. (**A**) Fv/Fm, the maximum photochemical efficiency of PSΠ and the efficiency of excitation energy capture by open PSΠ reaction centers, Fv′/Fm′. (**B**) Light energy distribution of PSΠ. [Y(Π)] was the actual quantum yield and Y(NPQ) and Y(NO) were the quantum yield of nonregulated and regulated energy dissipation, respectivley. (**C**) Non-photochemical quenching, NPQ; Photochemical quenching, qP. (**D**) The portion of energy reaction in PSII, where P is the portion of photochemical reaction, Ex is the portion of non-photochemical reaction, and D is the portion of antenna heat dissipation. Data are means ± SDs (n = 3). Different lowercase letters represent significant differences (*p* ≤ 0.05) in the same period among treatments, according to Duncan’s test.

**Figure 5 plants-12-02563-f005:**
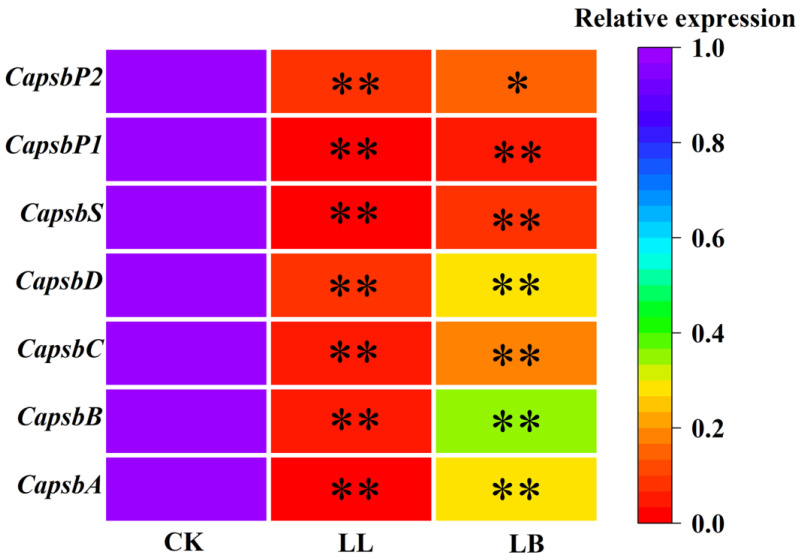
Expression patterns of differentially expressed genes related to the PSΠ reaction center proteins in pepper seedling leaves under LL stress for 7 days. The expression levels of the control group in the three treatment groups were used as the reference for the corresponding gene expression levels, and actin served as the internal standard. The figures show the expression levels of Capsb A, Capsb B, Capsb C, Capsb D, Capsb S, Capsb P1, and Capsb P2, respectively. The color scale corresponds to relative expression values. The more purple the block is, the higher the expression, and the more red, the lower the expression. Each row represents a unigene. CK, LL, and LB represent different treatments. Data are means ± SDs (n = 3). Different lowercase letters represent significant differences in the same period among treatments, according to Duncan’s test. the “*” represent significant differences (*p* ≤ 0.01), “**” represent significant differences (*p* ≤ 0.05).

**Figure 6 plants-12-02563-f006:**
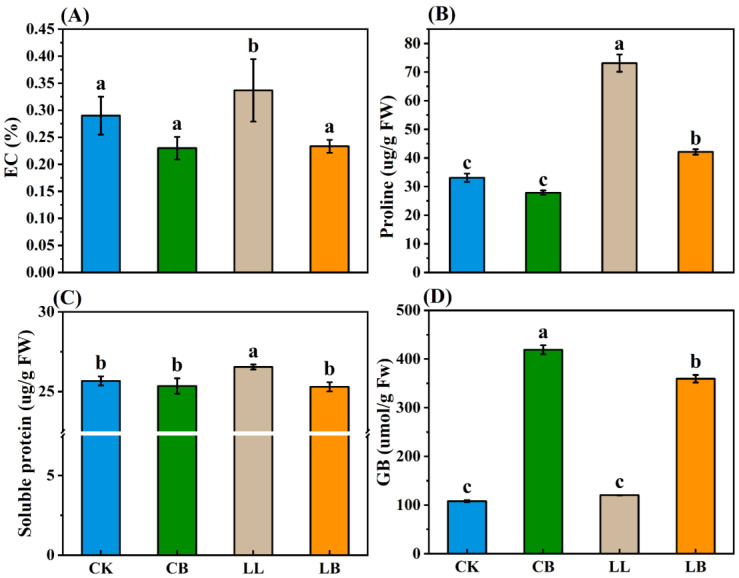
GB affected the osmotic substances by LL stress in pepper seedlings. Data of pepper plants under low-temperature conditions were obtained after 7 days. (**A**) EC: electrical conductivity. (**B**) Proline. (**C**) Soluble protein. (**D**) GB (glycine betaine) content. Different lowercase letters represent significant differences (*p* ≤ 0.05) in the same period among treatments, according to Duncan’s test.

**Figure 7 plants-12-02563-f007:**
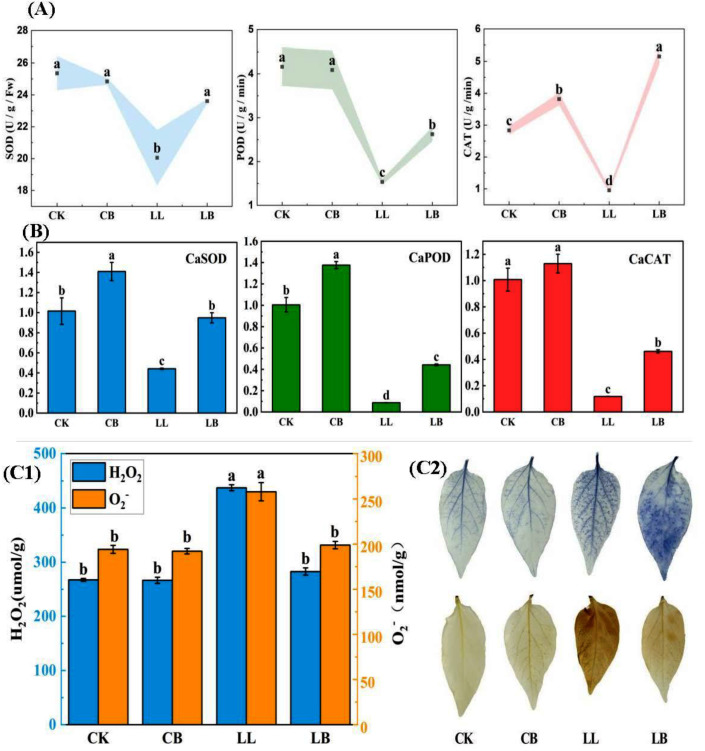
GB affected the antioxidant enzymes, relative expression, and histochemical staining by LL stress in pepper seedings. Data from pepper plants under low-temperature conditions were obtained after 7 days. (**A**) Contained SOD activity, POD activity, and CAT activity. (**B**) The relative expression of CaSOD, CaPOD, and CaCAT. (**C1**) H_2_O_2_ and O_2_^−^ content. (**C2**) NBT and DAB histochemical staining. Data are means ± SDs (n = 3). Different lowercase letters represent significant differences (*p* ≤ 0.05) in the same period among treatments, according to Duncan’s test.

**Figure 8 plants-12-02563-f008:**
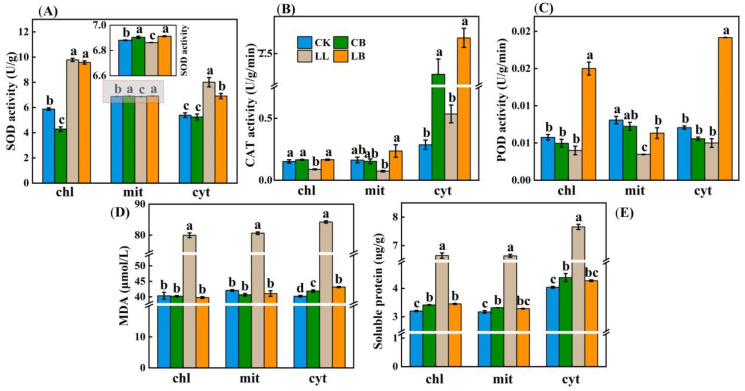
GB affected the subcellular localization of antioxidant systems in pepper leaves. Data from pepper plants under LL conditions were obtained after 7 days. (**A**) SOD activity in mitochondria (mit), chloroplast (chl), and cytosol (cyt). (**B**) CAT activity in mit, chl, and cyt. (**C**) POD activity in mit, chl, and cyt. (**D**) MDA content in mit, chl, and cyt. (**E**) Soluble protein content in mit, chl, and cyt. Data are means ± SDs (n = 3). Different lowercase letters represent significant differences (*p* ≤ 0.05) in the same period among treatments, according to Duncan’s test.

**Table 1 plants-12-02563-t001:** The sequences of primers used for the qRT-PCR.

Gene Name	Sequence (5′–3′)	GenBank Accession Number	Encoded Target	Amplicons Size (bp)
*Actin*	F: GTCCTTCCATCGTCCACAGG	XM_016722297.1	*Capsicum annuum* actin	1134
	R: GAAGGGCAAAGGTTCACAACA			
*CaSOD*	F: GTGAGCCTCCAAAGGGTTCTCTTG	AF036936.2	Manganese superoxide dismutase	687
	R: AAACCAAGCCACACCCAACCAG			
*CaPOD*	F: GCCAGGACAGCAAGCCAAGG	FJ596178.1	Peroxidase	975
	R: TGAGCACCTGATAAGGCAACCATG			
*CaCAT*	F: TTAACGCTCCCAAGTGTGCTCATC	NM_001324674.1	Catalase	1479
	R: GGCAGGACGACAAGGATCAAACC			
*CaPsbA*	F: GAATAGGGAGCCGCCGAATACAC	NC_018552.1:565–1626	Turnover proteins D1	1062
	R: TATTCCAGGCTGAGCACAACATCC			
*CaPsbB*	F: TGGGTTTGCCTTGGTATCGTGTTC	NC_018552.1:75670–77196	Internal light-harvesting complex proteins CP43	1527
	R: GCCCAACCAGCAACCAGAGC			
*CaPsbC*	F: GGATCTGCGTGCTCCATGGTTAG	NC_018552.1:35289–36674	Internal light-harvesting complex proteins CP47	1386
	R: CCGTTCCTGCCAAGGTTGTATGTC			
*CaPsbD*	F: TGGTCACCGCTAACCGCTTTTG	NC_018552.1:34244–35305	Turnover proteins D2	1062
	R: AGACCGACTACTCCAAGAGCACTC			
*CaPsbS*	F: AGGGAAAGGAGCATTGGCACAAC	XM_016719676.1	Photosystem II 22 kDa protein	834
	R: GCAGCAAAGAAGAAGAAGGCAACG			
*CaPsbP1*	F: GCTGCTTCCACACAATGCTTCTTG	XM_016701483.1	Oxygen-evolving enhancer protein 2	783
	R: TGGTTAGGCTTGAGGGTTGAAACG			
*CaPsbP2*	F: CTCGGGCAGCATTTGCTACCATAG	XM_016690370.1	*Capsicum annuum* photosynthetic NDH subunit	699
	R: CCTGAAATGAGTCGGCCACCAC			

Note: F: Forward primer; R: Revers primer.

## Data Availability

Data are contained within the article.

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
