# Peer review of "Foliar Spraying of Glycine Betaine Alleviated Growth Inhibition, Photoinhibition, and Oxidative Stress in Pepper (Capsicum annuum L.) Seedlings under Low Temperatures Combined with Low Light"

_plants, 2023, doi:10.3390/plants12132563_

Round 1
Reviewer 1 Report
The paper by Li et al shows that a foliar pre-treatment with 20 mM glycine betaine alleviates the negative effects of combined low temperature/low light stress. The experimental approach includes analysis of photosynthetic and photosynthesis-related parameters, measurements of growth and biomass production, quantification of osmolites, and evaluation of membrane function. Overall, all these parameters were better with GB treatment under low light/low temperature stress.
The subject is interesting for the general reader, and it is within the scope of the special issue “Horticultural Crops under Stresses”. The interest of the work has been accurately justified and the objectives are clear and correctly defined. Nevertheless, the manuscript could be improved by unifying and reducing redundant information. For example, data in 2.1 and 2.3, or 2.2 and 2.4, are redundant. There are also too many subsections. Indeed, one subsection for each Figure: subsections 2.1 to 2.9 for nine Figures.
Specific comments
Page 5, subsection 2.3. How has root activity been measured?
Page 9, subsection 2.7 and Figure 7. Explain the meaning of EC in the text and in Figure 8 legend. Explain how EC was measured.
Page 11, Figure 8. Follow the same order (SOD, CAT, POD) in Fig 8A and Fig 8B. The figure legend does not correspond to the content of Fig 8A, B, C1 and C2.
Discusion
-Subsections are not necessary
-Delete subsection 3.1
-A substantial part of the content of “discussion” is the same as in “results”
The manuscript needs a thorough revision of the language
Reviewer 2 Report
The manuscript deals with the evaluation of glycine betaine in mitigating low temperature and low light stress in pepper. The paper is interesting and combines assessment of plant biochemistry and genetics. It shows a lot of work put into the research. However, the paper has some flaws and requires English revision. The details are listed below:
L16-17: replace units in superscript. Correct throughout the paper
L22-31: add some % changes of examined parameters between treatments
L31: H2O2 and O2-. Correct throughout the paper
L48: Latin species and genus names in italics throughout the paper
L52-55: rephrase
L70-72: indicate the ubiquitous role of antioxidant enzymes in mitigating different types of stress (temperature, drought, pesticides, heavy metals). For this purpose the Authors may refer to the following reference: https://doi.org/10.1016/j.scienta.2022.110988
L102: ‘LL’ – indicate full name and abbreviation in brackets
L114: explain abbreviations
L138: the highest
L149-150: rephrase to reflect the meaning in the bars in Fig. 2
L160: rephrase
L162: ‘concluded’ – rephrase. It is a description of the results, not conclusion
L164: ;increased plant height’ – where are the results of plant height due to LB
L165: what do you mean by root activity?
L164-165: add % differences between parameters
L170: ‘on growth’ – on fresh and dry weight
L178-190: try to use the same abbreviations as in the figures, throughout the paper
L197: rephrase and explain the abbreviations
L261-207: explain the abbreviations used for the first time
L217, 275: spraying. Throughout the paper
L221: add units in the Fig. 5
L255-258: transfer this sentence to the discussion
L261-262: ‘EC’, ‘GB’ – indicate full name
L290-306: it was not mentioned in Materials and methods how cellular organelles were isolated and how these compounds were analyzed in each type of organelle
L293-294: it seems that SOD activity in mitochondria is not significant between treatments
L303: but the statistical significance letters are different
L307-308: it is not shown in Fig. 9. The activity of SOD in mitochondria seems to be comparable between treatments
L338: ;is not a bad approach’ – rephrase
L351: ‘Cr’ – full name
L369: ‘Cd’ – full name
L417: rephrase
L418: ‘consistent with previous studies’ – indicate the references
L420: ‘it was speculated’ – it was assumed
L452: at l LL – correct
L465: how many pots per treatment?
L471-476: indicate clearly that it was a preliminary experiment which was expanded in point 4.2.2
L503: ‘SW’ – explain
L513: tube
L554-555: (5 ml), (2 ml)
L555: remove one ‘supernatant’
L560-561: describe briefly
L577: correct the reference
L583-585: describe this method
L603: in Table 1 indicate which target is encoded by these genes and how many bp have the amplicons
L613: ‘LL’ – add full name of stress to conclude
L613-618: rephrase
Moderate editing of English language required
Round 2
Reviewer 2 Report
The manuscript has been significantly improved. I have no more comments.